# Self-assembled conjoined-cages

Sagarika Samantray[1], Shobhana Krishnaswamy[1] & Dillip K. Chand [1✉]

A self-assembled coordination cage usually possesses one well-defined three-dimensional (3D) cavity whereas infinite number of 3D-cavities are crafted in a designer metal-organic framework. Construction of a discrete coordination cage possessing multiple number of 3D-cavities is a challenging task. Here we report the peripheral decoration of a trinuclear $[Pd_3L_6]$ core with one, two and three units of a $[Pd_2L_4]$ entity for the preparation of multi-3D-cavity conjoined-cages of $[Pd_4(L^a)_2(L^b)_4]$, $[Pd_5(L^b)_4(L^c)_2]$ and $[Pd_6(L^c)_6]$ formulations, respectively. Formation of the tetranuclear and pentanuclear complexes is attributed to the favorable integrative self-sorting of the participating components. Cage-fusion reactions and ligand-displacement-induced cage-to-cage transformation reactions are carried out using appropriately chosen ligand components and cages prepared in this work. The smaller $[Pd_2L_4]$ cavity selectively binds one unit of $NO_3^-$, $F^-$, $Cl^-$ or $Br^-$ while the larger $[Pd_3L_6]$ cavity accommodates up to four DMSO molecules. Designing aspects of our conjoined-cages possess enough potential to inspire construction of exotic molecular architectures.

[1] Department of Chemistry, Indian Institute of Technology Madras, Chennai 600036, India. ✉email: dillip@iitm.ac.in

The construction of metal directed self-assembled complexes was pioneered by Lehn, Saalfrank, Sauvage, Fujita, Stang, Cotton, Raymond, Newkome, and others during the formative stages of this modern research area[1–9]. A diverse range of molecular architectures including, but not limited to, a variety of geometrical/topological shapes like macrocycles, cages, catenanes, and knots are known[10–13]. Self-assembled coordination macrocycles (monocyclic) typically possess one well-defined two-dimensional (2D) cavity whereas the coordination cages (polycyclic) usually possess one well-defined three-dimensional (3D) cavity[1–9]. Researchers have been actively exploring the synthesis of near-planar multi(monocyclic) coordination complexes that possess two or more 2D-cavities and the area is well-reviewed[5,14–16]. In contrast, coordination cages comprising two or more 3D-cavities are extremely rare and multi-compartmental in their builds[9,17–24]. Multi-compartment vesicles can facilitate individual chemical processes in distinct adjacent compartments[25]. Such behaviors are known in bio-systems as exemplified by functioning of regulatory mechanisms in prokaryotes[26]. Thus, the process of developing simpler routes for synthesizing multi-3D-cavity host molecules is a compelling and challenging task.

The early examples of multi-3D-cavity coordination cages were reported by Lehn and co-workers[17]. Initially, a specific Cu(I)-based box-shaped cationic $[M_6(L^a)_3(L^b)_2]$ cage containing a 3D-cavity was prepared using a linear bis-bidentate ligand ($L^a$) and trigonal planar tris-bidentate ligand ($L^b$)[27]. Subsequently, the linear ligand $L^a$ was modified by adding more binding units on its backbone. The modified $L^a$ being tris-bidentate and tetrakis-bidentate in nature, afforded $[M_9(L^a)_3(L^b)_3]$ and $[M_{12}(L^a)_3(L^b)_4]$ cages with two and three 3D-cavities, respectively[17]. The general formula of these cationic cages is $[M_{3n}(L^a)_3(L^b)_n]$ where M is Cu(I) or Ag(I), $n$ is 2, 3, or 4 and the number of cavities is "$n$ – 1". Schmittel et al.[18] prepared another Cu(I)-based $[M_6(L^a)_3(L^b)_2]$ cage and then subjected the three bound ligands $L^a$ to post-modification using two units of a tripodal linker to obtain one unit of a bound macrobicyclic cyclophane entity. Thus, a $[M_6(cyclophane)(L^b)_2]$ type cationic compound, containing three 3D-cavities was prepared. Hardie and co-workers prepared a Cu(II)-based neutral dumbbell-shaped $[M_3(L^a)_2(dmf)_3]$ cage where $L^a$ represents a tri-anionic tripodal tris-monodentate ligand. Two units of the trinuclear cage were linked using a neutral bis-monodentate linear linker ($L^b$), to afford a neutral $[\{M_3(L^a)_2(dmf)(H_2O)\}_2(\mu\text{-}L^b)]$ architecture that contains identical 3D-cavities[19]. A few years ago, we prepared a Pd(II)-based cationic double-decker $[M_3L_4]$ cage possessing two identical 3D-cavities (Fig. 1a)[20]. We prepared the $[M_3L_4]$ cage using $Pd(NO_3)_2$ and an "E-shaped" neutral tris-monodentate ligand in 3:4 ratio. Our design enables the creation of tuneable cavities by keeping the donor units of the ligand intact and simply modifying the spacer moieties. A few other $[Pd_3L_4]$ complexes were subsequently reported by the research groups of Clever, Yoshizawa and Crowley[21–23], by suitably modifying the spacer units in the ligand backbones to realize bigger sized $[Pd_3L_4]$ double-cavities. The Crowley group introduced an additional donor site in the ligand design, creating a tetrakis-monodentate ligand that allowed the formation of a $[Pd_4L_4]$ complex with three 3D-cavities arranged in a linear fashion[23]. The environment of central cavity, by virtue of its position, has to be different from a terminal cavity, however, there are subtle differences in the frameworks of the central versus terminal cavities in the design of Crowley. In short, a few examples of $[M_nL_4]$ multi-cavity cages (where M is Pd(II), "$n$" is 3 or 4) with "$n – 1$" cavities (Fig. 1a) are known. In contrast to the multiple binding sites of the multi-3D-cavity cages, there exist single-3D-cavity systems capable of accommodating multiple variety of guests in site-specific manner[28,29]. It is pertinent to note that a variety of 3D-metal—organic frameworks possessing an infinite number of conjoined-cages in their architectures are known[7]. As described above, only a handful of multi-3D-cavity self-assembled cages are known in literature, where the cavities are usually arranged in a linear fashion in their superstructures and the maximum number of cavities is three.

The processes of construction of coordination cages are sometime classified under narcissistic and integrative self-sorting[30,31] that are comparable to certain biological processes[32–34]. Dynamic behaviors of the coordination cages with respect to post-modifications such as cage-fusion reactions[35] and ligand-displacement-induced cage-to-cage transformations[36] are important studies of current interest. Construction of multi-3D-cavity cages, understanding related self-sorting processes and subjecting the cages to post-modifications are therefore attractive and fundamental aspects of supramolecular coordination chemistry.

In the present work, we report a family of rationally designed modular multi-cavity Pd(II)-based coordination cages where 3D-cages of two varieties, namely $[Pd_2L_4]$ and $[Pd_3L_6]$ are conjoined in a linear or lateral manner (Fig. 1b). We have named such architectures as "self-assembled conjoined-cages". A family of conjoined 3D-cages of $[Pd_4(L^a)_2(L^b)_4]$, $[Pd_5(L^b)_4(L^c)_2]$ and $[Pd_6(L^c)_6]$ formulations containing two, three, and four cavities, respectively, has been prepared (Fig. 1b). The smaller peripheral cavity of the conjoined-cages selectively binds certain anions while the larger central cavity contains solvent molecules. Dynamic behaviors of the conjoined-cages depicting cage-fusion reactions and ligand-displacement-induced cage-to-cage transformations are studied. Narcissistic and integrative self-sorting processes are demonstrated in connection with the synthesis of the cages.

## Results

**Design and synthesis of the ligands.** The complexation of Pd(II) with suitable nonchelating bidentate ligands is known to yield $[Pd_mL_{2m}]$ complexes. The value of "$m$" can be qualitatively

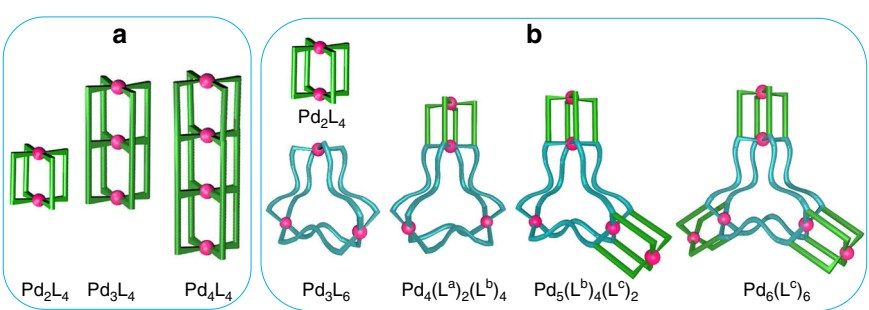

**Fig. 1 Cartoon representation of the design approaches for making of self-assembled multi-3D-cavity cages. a** linearly conjoined homoleptic cages[20,23] and **b** linearly/laterally conjoined targeted homo- and heteroleptic cages (this work).

**Fig. 2 Structure of the ligands.** The ligands **L1–L6**.

related to the angle subtended by the two coordination vectors of the bound ligand. While [Pd₂L₄] complexes are common and have been widely explored[6,37–39], molecules with [Pd₃L₆], [Pd₄L₈], [Pd₅L₁₀], [Pd₆L₁₂], [Pd₇L₁₄], [Pd₈L₁₆], [Pd₉L₁₈], [Pd₁₂L₂₄], [Pd₃₀L₆₀], [Pd₄₈L₉₆] architectures possessing a single-3D-cavity are also known[6,40–42]. In order to accomplish the multi-3D-cavity targets shown in Fig. 1b, the first step was to identify two nonchelating bidentate ligands; one capable of forming a [Pd₂L₄] and the other a [Pd₃L₆] complex. Mere identification of any two capable ligands is not sufficient, since the backbones of the chosen ligands need to be integrated in such a manner that the hybrid ligands so obtained can sustain [Pd₂L₄] and [Pd₃L₆] entities within the same superstructure. The objectives include the construction of tetra, penta and hexanuclear complexes shown in Fig. 1b. The ligands designed for this purpose are shown in Fig. 2. The bidentate ligands **L2** and **L3** yielded [Pd₂L₄] and [Pd₃L₆] architectures, respectively. The builds of **L2** and **L3** are integrated in the designs of the tri-/tetradentate ligands **L5/L6**.

The ligand **L1** was prepared as reported[20] and **L2** by a modified method[43]. The new ligands **L3–L6** were synthesized as described hereafter. The ligands **L2** and **L3** were obtained by condensation of nicotinoyl chloride hydrochloride with 3-pyridylcarbinol and resorcinol, respectively. The ligand **L4** was obtained by selective cleavage of one of the ester linkages of **L1**, whereas selective condensation of nicotinic acid with resorcinol resulted in the ligand **L4′**. The ligand **L5** was synthesized by condensation of **L4** with **L4′**, whereas the ligand **L6** was prepared by condensation of **L4** with resorcinol. The ligands were characterized by nuclear magnetic resonance (NMR) spectroscopy and electrospray ionization mass spectrometry (ESI-MS) techniques. Ligand **L1** upon complexation with Pd(NO₃)₂ forms a double-decker architecture [(NO₃)₂ ⊂ Pd₃(**L1**)₄](NO₃)₄, **1a**[20].

**[Pd₂L₄] entity**. Complexation of Pd(NO₃)₂ with the ligand **L2** in 1:2 ratio was carried out in dimethyl sulfoxide (DMSO)-$d_6$. Spontaneous assembly of the components resulted in the complex [NO₃ ⊂ Pd₂(**L2**)₄](NO₃)₃, **2a** within 10 min at room temperature (Fig. 3a). The ¹H NMR spectrum of the solution (Fig. 4a) contained multiple sets of signals showing complexation induced downfield shift for relevant protons. Multiple sets of signals are typically associated with either the existence of a dynamic

equilibrium of two or more complexes[44] of different formulations or a single complex[44] with multiple isomers where the environment around the bound ligand units differs in each isomer. The ligand **L2**, being unsymmetrical, can exist in two possible orientations when bridged between two metal centers. Consequently, four isomeric molecular architectures of [Pd₂L₄] composition (diastereomers) differing in the relative orientations of the bound ligand units are possible. While a statistical mixture of diastereomers in **2a** was supported by the ¹H NMR spectrum of the sample, the [Pd₂L₄] composition was proposed based on ESI-MS studies. The addition of one equivalent of TBAX (tetra-n-butylammonium salts) (X = F⁻, Cl⁻ or Br⁻) to a solution of **2a** resulted in the corresponding anion exchanged products [X ⊂ Pd₂(**L2**)₄](NO₃)₃, **2b–2d** within 5 min at 70 °C (Fig. 3b), which exhibited downfield shift of pyridine-α protons. The presence of a NO₃⁻ ion in the cavity and its templating role was further supported by the fact that complexation of Pd(BF₄)₂ with the ligand **L2** in 1:2 ratio provided a mixture of several unidentified products. This mixture could be converted to [X ⊂ Pd₂(**L2**)₄](BF₄)₃, **2a′-2d′** (X = NO₃⁻, F⁻, Cl⁻, and Br⁻, respectively), by the addition of one equivalent of the corresponding TBAX, within 5 min at 70 °C. The representative complexes **2b** and **2a′** were also characterized by ESI-MS studies. The crystal structure of **2c** supported the [Pd₂L₄] architecture (Fig. 5a) with an encapsulated Cl⁻ ion. The two possible orientations for each ligand strand perhaps introduce partial occupancies for –C(O)– and –CH₂– at both ends of the –C(O)OCH₂– spacer moiety. The occupancies could not be resolved properly, and the crystal structure represents a mixture of the four isomeric complexes.

**[Pd₃L₆] entity**. The complexation of Pd(NO₃)₂ with the ligand **L3** in 1:2 ratio was carried out in DMSO-$d_6$. Spontaneous assembly of the components resulted in the complex [Pd₃(**L3**)₆](NO₃)₆, **3a** within 10 min at room temperature (Fig. 3c) and the ¹H NMR spectrum of the solution (Fig. 4b) showed a single set of signals where a downfield shift was seen in the positions of the pyridine-α protons. The [Pd₃L₆] composition of **3a** was proposed based on ESI-MS studies. Synthesis of the complexes [Pd₃(**L3**)₆](X)₆, **3b–3f** (for X = BF₄⁻, ClO₄⁻, OTf⁻, PF₆⁻ and SbF₆⁻, respectively) was completed within 10 min at room temperature and the ¹H NMR spectra of these complexes are all comparable (except for

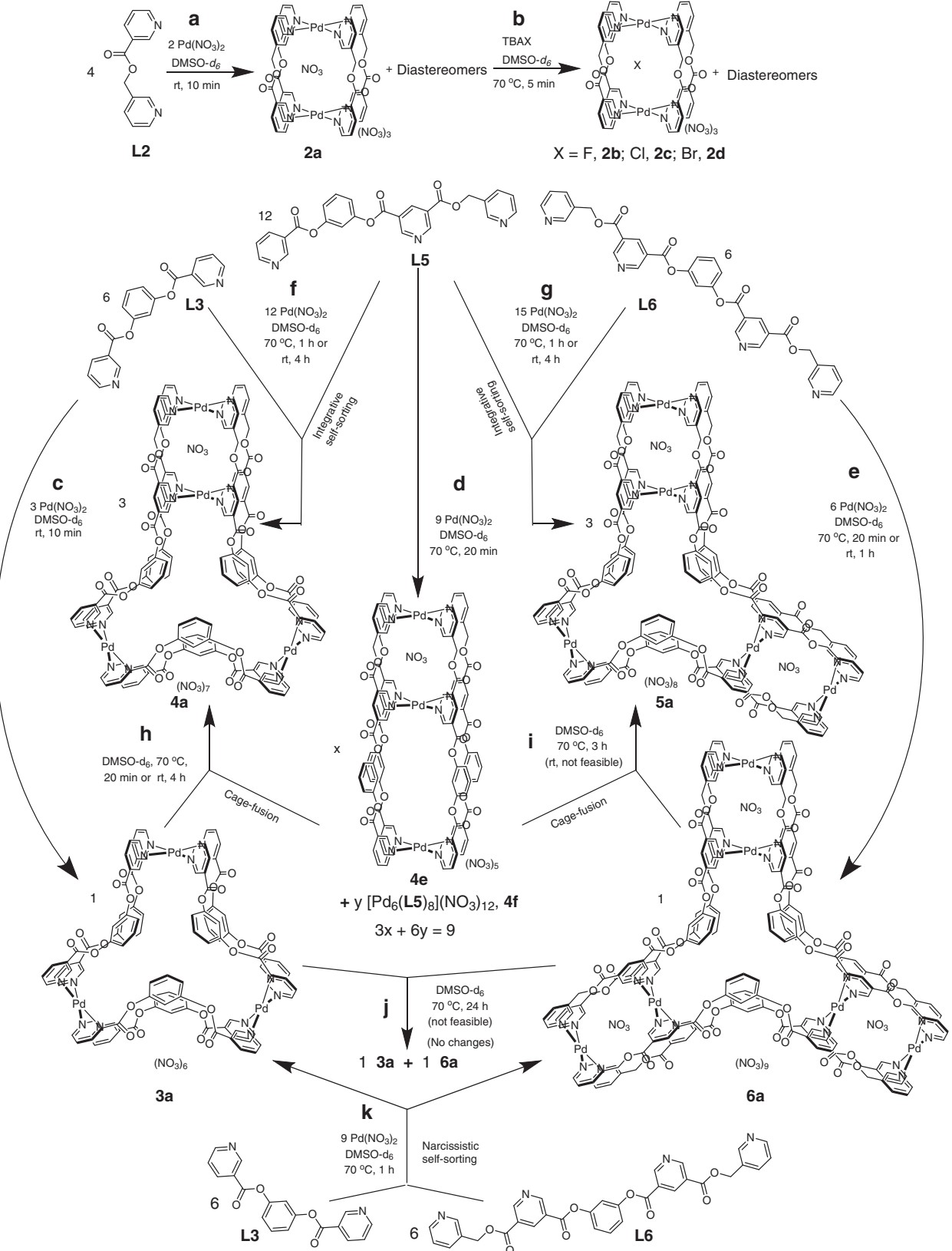

**Fig. 3 Synthetic scheme for the complexes 2a–2d, 3a–6a, and 4e/4f.** Self-assembled coordination cages featuring one or more 3D-cavity constructed by complexation of $Pd(NO_3)_2$ with appropriate ligand(s) to afford **a/b/c/d/e** homoleptic complexes; **f/g** heteroleptic complexes via integrative self-sorting; **k** mixture of homoleptic complexes via narcissistic self-sorting. Cage-fusion reactions to yield **h/i** heteroleptic complexes (however, no fusion in the case of **j**).

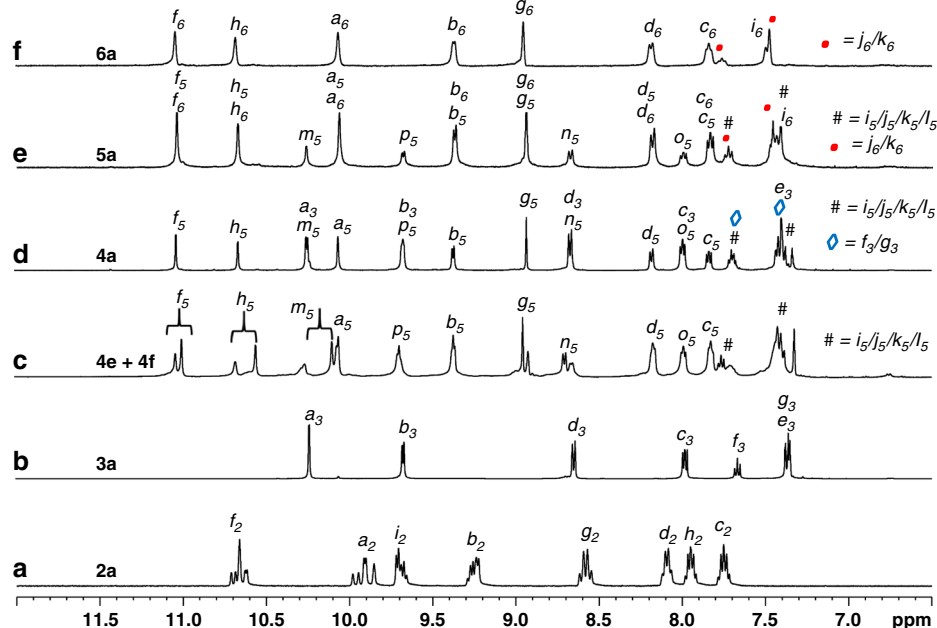

**Fig. 4 Characterization of the complexes 2a−6a and 4e/4f.** Partial $^1$H NMR spectra (400 MHz, DMSO-$d_6$, 300 K) of **a** cage **2a** (diastereomeric mixture), **b** cage **3a** (trinuclear), **c** mixture of **4e** and **4f** (tri- and hexanuclear), **d** cage **4a** (tetranuclear), **e** cage **5a** (pentanuclear), and **f** cage **6a** (hexanuclear).

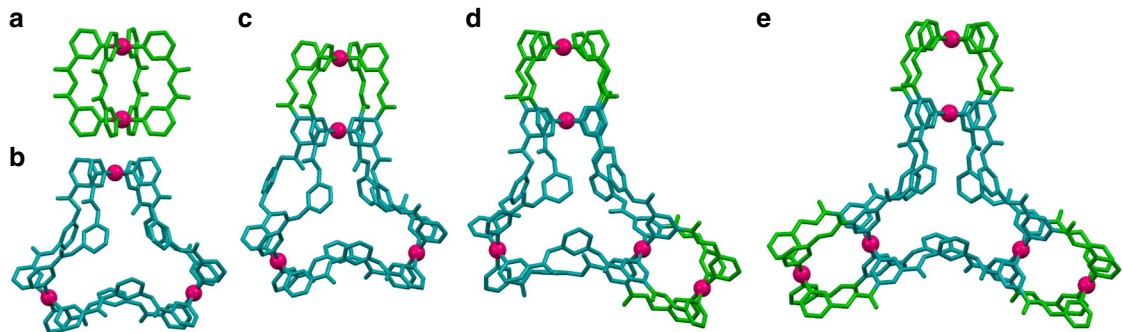

**Fig. 5 Crystal structures showing the cationic portions. a** Cage **2c** (binuclear), **b** cage **3a** (trinuclear), **c** cage **4acl** (tetranuclear), **d** cage **5c** (pentanuclear), and **e** cage **6c** (hexanuclear) (encapsulated guests, counter-anions, solvents, and hydrogen atoms are excluded for clarity. *ORTEP* diagram for complexes and suitable crystal structures showing encapsulated guests are available in the Supplementary Figs. 134–143).

minor differences in the signal of H$_{g3}$). Thus, the role of anion templation in the formation of these trinuclear complexes (**3a**–**3f**) was ruled out. However, encapsulation of solvent molecules in the cavity of the complexes is likely. ESI-MS study of the representative complex [Pd$_3$(**L3**)$_6$](BF$_4$)$_6$, **3b** supported the [Pd$_3$L$_6$] composition.

$^1$H NMR spectra of the complex **3a** were recorded at various concentrations. At a very low concentration (1 mM with respect to Pd(II)) the complex **3a** started dissociating, releasing approximately 19% of uncoordinated ligand **L3**. Interestingly, ~7% of **L3** existed as a binuclear complex [Pd$_2$(**L3**)$_4$](NO$_3$)$_4$, **3g**, of the [Pd$_2$L$_4$] variety, while the trinuclear **3a** remained the major species. At higher concentration, (30 mM with respect to Pd(II)) approximately ~6% of **L3** existed as a tetranuclear complex [Pd$_4$(**L3**)$_8$](NO$_3$)$_8$, **3h**, of the [Pd$_4$L$_8$] variety, whereas the trinuclear **3a** remained the major species (see Supplementary Fig. 30). The evolution of the smaller **3g** and larger sized **3h** (Supplementary Discussion 1 and Supplementary Table 1) at lower and higher concentrations, respectively, were proposed based on entropic concepts[44]. The formation of **3g** and **3h** was also confirmed using ESI-MS data (see Supplementary Figs. 31, 32). Attempts to grow single crystals of these complexes proved unsuccessful. PM6

optimized structures of **3g** and **3h** are given in lieu of the crystal structures (see Supplementary Fig. 132).

The bidentate ligand **L3** possesses a central aromatic spacer and two terminal 3-pyridyl moieties connected by ester linkages. A few ligands of comparable designs with amide linkages are known, which form [Pd$_2$L$_4$] complexes[45–47]. The amide linkages are somewhat rigid and are capable of interacting with counter-anions inside the corresponding cavity, when suitably oriented thereby influencing the formation of smaller [Pd$_2$L$_4$] complexes. The observed strong preference of **L3** towards the formation of a [Pd$_3$L$_6$] complex was rather surprising. Probably, the ester linkages are not suitable for anion binding and their flexible nature allows conformational changes when required. In any case, we needed a ligand, which would yield a [Pd$_3$L$_6$] architecture, regardless of the counter-anion present, within a reasonable concentration range. The ligand **L3** fits this requirement and satisfies few other criteria necessary to achieve the targets (shown in Fig. 1b). The crystal structure of the complex **3a** revealed a bent conformation of the bound ligand moieties, where the donor atoms are present at the convex face of the curved ligand (Fig. 5b). Four DMSO molecules are located inside the cavity and the counter-anions are present outside.

**Conjoined-cages and differential binding**. The next target was a $[Pd_4(L^a)_2(L^b)_4]$ complex that can be visualized as a linear conjoin of a $[Pd_2L_4]$ cage with a $[Pd_3L_6]$ cage. Therefore, complexation of $Pd(NO_3)_2$ (4 equiv.) with a mixture of the ligands L3 (2 equiv.) and L5 (4 equiv.) was carried out in DMSO-$d_6$. Integrative self-sorting of these components required 4 h at room temperature or 1 h at 70 °C, as revealed by monitoring of $^1H$ NMR spectra, yielding $[NO_3 \subset Pd_4(L3)_2(L5)_4](NO_3)_7$, **4a** (Fig. 3f). The $^1H$ NMR spectrum of the solution (Fig. 4d) showed a single set of peaks where a downfield shift was seen in the positions of the pyridine-α protons. Addition of TBAX ($X = F^-$, $Cl^-$, or $Br^-$) to a solution of **4a** resulted in the corresponding anion exchanged products $[X \subset Pd_4(L3)_2(L5)_4](NO_3)_7$, **4b–4d** (for $X = F^-$, $Cl^-$, and $Br^-$, respectively) within 5 min at 70 °C. Addition of AgCl to a solution of **4a** took longer time for the complete anion exchange when carried out at room temperature, however, at an initial stage partial anion exchange was observed (a mixture of **4a** and **4c**) as confirmed by $^1H$ NMR study. $^1H$ NMR spectra of the solution recorded at an intermediate stage revealed the presence of a mixture of **4a** and **4c**. Such a mixture was used for growing single crystals and crystals were obtained from two of the crystallization conditions. Crystal structures obtained from both the samples displayed partial occupancies of encapsulated $NO_3^-/Cl^-$ ion. The crystal structure of **4acI**, revealed the formation of a tetranuclear complex where two cavities are linearly conjoined (Fig. 5c). The smaller cavity accommodated a $NO_3^-/Cl^-$ ion (with partial occupancies) and four DMSO molecules were present inside the bigger cavity. The counter-anions and a few solvent molecules were located outside the cavities. The crystal structure of **4acII** is provided in the Supplementary Information.

As explained earlier, the complexation of $Pd(NO_3)_2$ with ligand L3 yielded the homoleptic complex **3a**. It is also relevant to discuss the complexation behavior of $Pd(NO_3)_2$ with ligand L5. Since the ligand L5 structurally resembles a combination of L2 and L3, hence the binding sites of L5 are suited for making $[Pd_2L_4]$ and $[Pd_3L_6]$ entities. Therefore, we pondered reasonable architectures where all three donor sites of L5 and all four acceptor sites around Pd(II) are completely utilized and the anticipated $[Pd_2L_4]$ and $[Pd_3L_6]$ like entities are sustained. A structure could not be readily visualized, nevertheless, complexation of $Pd(NO_3)_2$ with ligand L5 in 3:4 ratio was performed in DMSO-$d_6$ by stirring the mixture for 1 h at 70 °C (Fig. 3d). $^1H$ NMR spectrum of the solution (Fig. 4c) exhibited two sets of signals, which appear downfield relative to the corresponding ligand protons. The $^1H$, $^{13}C$, H-HCOSY, and NOESY NMR data along with description about the complexes are given in supplementary section (see Supplementary Figs. 96–100 and Supplementary Discussions 2–4). While the $^1H$ NMR spectra recorded at different temperatures (30 to 100 °C range) (see Supplementary Fig. 101) did not show any noticeable change those recorded at different concentrations (see Supplementary Fig. 102) showed changes in the relative intensities of the signals, indicating the coexistence of two well-defined complexes. Their architectures could not be readily predicted. ESI-MS data provided evidence to propose the formation of $[NO_3 \subset Pd_3(L5)_4](NO_3)_5$, **4e** and $[(NO_3)_2 \subset Pd_6(L5)_8](NO_3)_{10}$, **4f** (see Supplementary Fig. 103). The structures of **4e** and **4f** are such that the L2-like fragment of L5 got manifested in the $[Pd_2L_4]$ form (in both **4e** and **4f**), whereas the L3-like fragment of L5 evolved in the $[Pd_2L_4]$ (in **4e**) and $[Pd_4L_8]$ (in **4f**) forms (see Supplementary Fig. 95). This observation is in line with the fact that the complexation of $Pd(NO_3)_2$ with ligand L3 resulted in very small proportions of $[Pd_2(L3)_4](NO_3)_4$, **3g** and $[Pd_4(L3)_8](NO_3)_8$, **3h** at low and high concentrations, respectively. Addition of AgCl to a mixture of **4e** and **4f** resulted in the anion exchanged products $[Cl \subset Pd_3(L5)_4](NO_3)_5$, **4g** and $[(Cl)_2 \subset Pd_6(L5)_8](NO_3)_{10}$, **4h**

within 30 min at room temperature (see Supplementary Fig. 104). The compositions of **4g** and **4h** were also supported by ESI-MS data (see Supplementary Fig. 105). Attempts to grow single crystals of these complexes proved unsuccessful. PM6 optimized structures of **4g** and **4h** are given in lieu of the crystal structures (see Supplementary Fig. 133).

A $[Pd_5(L^b)_4(L^c)_2]$ type complex that approximates a lateral conjoining of two $[Pd_2L_4]$ cavities around a $[Pd_3L_6]$ core was our next target. The complexation of $Pd(NO_3)_2$ (5 equiv.) with a mixture of the ligands L5 (4 equiv.) and L6 (2 equiv.) was carried out in DMSO-$d_6$ in anticipation of the integrative self-sorting behavior of the system (Fig. 3g). The self-sorting of the components occurred within 4 h at room temperature (or 1 h at 70 °C), as revealed by $^1H$ NMR study, yielding $[(NO_3)_2 \subset Pd_5(L5)_4(L6)_2](NO_3)_8$, **5a**. The $^1H$ NMR spectrum of the solution (Fig. 4e) showed a single set of peaks where pyridine-α proton signals appear downfield relative to those of the ligands. The composition of **5a** was proposed based on ESI-MS data. Addition of TBAX ($X = F$, $Cl$, or $Br$) to a solution of **5a** resulted in the corresponding anion exchanged products $[(X)_2 \subset Pd_5(L5)_4(L6)_2](NO_3)_8$, **5b–5d** (for $X = F^-$, $Cl^-$, and $Br^-$, respectively) within 5 min at 70 °C. The crystal structure of the complex **5c** revealed laterally conjoined cavities as anticipated (Fig. 5d). The two smaller cavities accommodated a $Cl^-$ ion each and four DMSO molecules were present inside the larger cavity. The counter-anions and a few solvent molecules were located outside the cavities.

The complexation of $Pd(NO_3)_2$ with the ligand L6 in 1:1 ratio was carried out in DMSO-$d_6$ (Fig. 3e). Spontaneous assembly of the components resulted in the complex $[(NO_3)_3Pd_6(L6)_6](NO_3)_9$, **6a** within 1 h at room temperature or 20 min at 70 °C. The $^1H$ NMR spectrum of the solution (Fig. 4f) showed a single set of signals where the peaks of pyridine-α protons showed a downfield shift. The $[Pd_6L_6]$ composition of **6a** was proposed based on ESI-MS data. Since the ligand L6 structurally resembles a combination of L2 and L3, complexation of Pd(II) with L6 affords the targeted $[Pd_6(L^c)_6]$ complex. Thus, our objective of synthesizing a complex containing three $[Pd_2L_4]$ cavities laterally conjoined with a $[Pd_3L_6]$ core was successfully accomplished.

Addition of TBAX ($X = F^-$, $Cl^-$, or $Br^-$) to a solution of **6a** resulted in the corresponding anion exchanged products $[(X)_3 \subset Pd_6(L6)_6](NO_3)_9$, **6b–6d** (for $X = F^-$, $Cl^-$, and $Br^-$, respectively) within 5 min at 70 °C. The composition of **6a** and **6c** were also supported by ESI-MS data. The crystal structure of the complex **6c** revealed laterally conjoined cavities as anticipated (Fig. 5e). The three smaller cavities accommodated a $Cl^-$ ion each and three DMSO molecules were present inside the larger cavity. The counter-anions and a few solvent molecules were located outside the cavities.

It was interesting to note that a combination of $Pd(BF_4)_2$, L3 and L5 in 4:2:4 ratio resulted in a mixture of several unidentified products. The mixture of products could, however, be converted to $[X \subset Pd_4(L3)_2(L5)_4](BF_4)_7$, **4a′–4d′** ($X = NO_3^-$, $F^-$, $Cl^-$, and $Br^-$, respectively) by addition of the corresponding TBAX and the process was complete in 1 h at 70 °C. Similarly, a mixture of products was obtained when (i) $Pd(BF_4)_2$, L5 and L6 were combined in 5:4:2 ratio or (ii) $Pd(BF_4)_2$ and L6 were combined in 1:1 ratio. These mixtures could be converted to discrete (i) $[(X)_2 \subset Pd_5(L5)_4(L6)_2](BF_4)_8$, **5a′–5d′** and (ii) $[(X)_3 \subset Pd_6(L6)_6](BF_4)_9$, **6a′–6d′** complexes via the addition of TBAX ($X = NO_3^-$, $F^-$, $Cl^-$, or $Br^-$).

**Cage-fusion reactions**. Cage-fusion reactions were investigated by combining any two cages, from the pool of **3a**, **4a**, **5a**, **6a**, and **4e**. Although **4e** and **4f** coexist, for ease of understanding and

calculation, the presence of **4f** was neglected. The cage-fusion reactions were monitored by recording $^1H$ NMR spectra of the solutions as a function of time. The combination of the two homoleptic systems **3a** and **4e** in 1:3 ratio in DMSO-$d_6$ resulted in the heteroleptic system **4a** within 4 h at room temperature or 20 min at 70 °C (Fig. 3h). In another instance, the combination of the homoleptic systems **6a** and **4e** in 1:3 ratio in DMSO-$d_6$ resulted in the heteroleptic system **5a**, within 4 h at 70 °C (no changes occurred at room temperature) as shown in Fig. 3i. The **L3**-like fragment might prefer to form a [Pd$_3$L$_6$] entity, however, this fragment in the complex **4e** exists in the less preferred [Pd$_2$L$_4$] form. Consumption of **4e** and the formation of **4a** or **5a** containing the preferred [Pd$_3$L$_6$] entity is considered as the driving force of the cage-fusion reactions. However, a mixture of the homoleptic complexes **3a** and **6a** remained unchanged even after stirring for 24 h at 70 °C (Fig. 3j). Presumably, the complex **6a** is quite stable and requires higher energy for a reshuffle in its architecture. No cage-fusion was observed when the heteroleptic complexes **4a** and **5a** were allowed to interact with each other. Similarly, no fusion was observed in experiments involving a homoleptic-heteroleptic pair of complexes such as **3a/4a**, **3a/5a**, **4e/4a**, **4e/5a**, **6a/4a**, and **6a/5a** (see Supplementary Methods and Supplementary Figs. 106–117, for all the cage-fusion reactions).

The integrative self-sorting phenomenon could be demonstrated in terms of the synthesis of **4a** and **5a** in separate reactions using Pd(NO$_3$)$_2$ and appropriate ligands as shown in Fig. 3f, g. These two integrative self-sorted complexes could also be prepared by cage-fusion reactions as discussed above. Thus, a cage-fusion reaction or direct combination of corresponding metal and ligand components yield the same final product, presumably through different routes[30]. An unsuccessful cage-fusion reaction (or no change) belongs in the category of narcissistic self-sorting. One such example of narcissistic self-sorting is observed when Pd(NO$_3$)$_2$ is mixed with the ligands **L3** and **L6** in one-pot (Fig. 3k and Supplementary Fig. 110), yielding a mixture of the corresponding homoleptic complexes **3a** and **6a** only.

**Ligand-displacement-induced cage-to-cage transformations.** Subsequently, a variety of ligand-displacement-induced cage-to-cage transformations were attempted as shown in Fig. 6. A chosen cage was mixed with a calculated amount of externally added ligand(s), whereupon the bound ligand(s) are partially or completely displaced by the incoming ligand(s), leading to complete disappearance of the original cage and formation of a different cage. For example, the cage **3a** could be transformed to the cage **6a** in a cage-to-cage fashion via the interaction of **3a** (2 equiv.) with **L6** (6 equiv.) whereupon **L3** (12 equiv.) and **6a** (1 equiv.) were obtained (Fig. 6b). The list of successful cage-to-cage transformations include the conversion of **3a** to **4a**, **5a** or **6a** (Fig. 6a, g, b); conversion of **4e** to **4a** or **6a** (Fig. 6e, f); conversion of **4a** to **6a** (Fig. 6c); and conversion of **5a** to **6a** (Fig. 6d). The cage-to-cage transformations were monitored by recording $^1H$ NMR spectra of the solutions as a function of time. All these transformations were complete in about 1 h at 70 °C. Fourteen different combinations were tried out of which seven (mentioned above) were successful (see Supplementary Methods and Supplementary Figs. 118–131, for all ligand-displacement reactions). The cages produced are probably more stable than the reactant cages in a qualitative sense.

## Discussion

This article demonstrated the construction of multi-3D-cavity coordination cages via decoration of a [Pd$_3$L$_6$] core with one or more [Pd$_2$L$_4$] units in a linear or lateral fashion, respectively. The

metal component used for the preparation of the cages was Pd(NO$_3$)$_2$ and the cages formed (**2a**, **4a**, **5a**, and **6a**) were found to encapsulate NO$_3^-$ in their [Pd$_2$L$_4$] moieties. The encapsulated NO$_3^-$ could be replaced by halides like F$^-$, Cl$^-$, or Br$^-$ by using corresponding TBAX. Notably, AgCl could be also used as a source of Cl$^-$ ion. In fact, Clever and co-workers[48] used sparingly soluble AgCl as a source of Cl$^-$ that displaced bound BF$_4^-$ ion from the cavity of certain coordination cages, resulting in consumption of AgCl and retention of the more soluble AgBF$_4$ in solution. We have also demonstrated the use of AgCl where the Cl$^-$ ion displaced bound NO$_3^-$ ion from the cavity of some coordination cages whereupon the more soluble AgNO$_3$ remained in solution[49]. The requirement of the encapsulated anions in the creation of these assemblies was realized when attempts toward synthesizing Pd$_2$L$_4$, [Pd$_4$(L$^a$)$_2$(L$^b$)$_4$], [Pd$_5$(L$^b$)$_4$(L$^c$)$_2$], and [Pd$_6$(L$^c$)$_6$] cages using Pd(BF$_4$)$_2$ failed and the experiments led to the formation of a mixture of unidentified products. Probably, the formation of the [Pd$_2$L$_4$] entity was hindered, due to repulsion between the closely placed metal ions, in the absence of an appropriate anionic template. This hindrance in turn prevented the building of the targeted conjoined-cages. The BF$_4^-$ ion was found to be un-suitable as a template here. The addition of TBAX (where X = NO$_3^-$, F$^-$, Cl$^-$, and Br$^-$, respectively) to these mixtures yielded the desired cages. Hence, the formation of Pd$_2$L$_4$, [Pd$_4$(L$^a$)$_2$(L$^b$)$_4$], [Pd$_5$(L$^b$)$_4$(L$^c$)$_2$], and [Pd$_6$(L$^c$)$_6$] cages is feasible only when the smaller cavity is occupied by NO$_3^-$, F$^-$, Cl$^-$, or Br$^-$, irrespective of the counter-anion present outside the cavity/cavities. Although the formation of the larger cavity (i.e., [Pd$_3$L$_6$] entity) is anion independent, this did not help in the formation of corresponding conjoined-cages (**4a**, **5a**, and **6a**) since the formation of [Pd$_2$L$_4$] entity is essential.

Synthesis of the heteroleptic complex **4a** through the combination of Pd(NO$_3$)$_2$, **L3** and **L5** in a single-pot is a perfect example of integrative self-sorting behavior[30,31] since the homoleptic complexes formed by **L3** and **L5** were not observed in the final product profile. Integrative self-sorting was also observed for synthesis of the heteroleptic complex **5a** from its components, i.e., Pd(NO$_3$)$_2$, **L5** and **L6**. The formation of these heteroleptic complex is probably driven by the propensity of the **L3**-like fragment present in **L5** (and **L6**) to form [Pd$_3$L$_6$]-like entities.

The cages **4a** and **5a** are hitherto unknown examples of heteroleptic complexes wherein ligands of different denticity are coordinated to Pd(II)[8]. Thus, the present set of cages are unique due to their structural features and associated binding properties. The concept of making conjoined-cages, in an effortless manner opens a plethora of possibilities where suitable ligand design can help construct hitherto unknown architectures with unique structures.

## Methods

**General.** The deuterated solvent DMSO-$d_6$ was obtained from Sigma-Aldrich. NMR spectra were recorded in DMSO-$d_6$ at room temperature (r.t.) on Bruker AV400 and AV500 spectrometers at 400 and 500 MHz for $^1H$ NMR, COSY, NOESY and at 100 and 125 MHz for $^{13}C$ NMR. Chemical shifts are reported in parts per million (ppm) relative to residual solvent protons (2.50 ppm for DMSO-$d_6$ in $^1H$ NMR and 39.50 in $^{13}C$ NMR). The ESI mass spectra were recorded on Agilent Q-TOF spectrometers. Single crystal X-ray diffraction analysis was carried out using a Bruker D8 VENTURE instrument. The ligand **L1** was synthesized as reported previously[20].

**Synthesis and characterization of the ligands.** Triethylamine (0.24 mL, 1.685 mmol) was added in a dropwise manner to a stirred suspension of nicotinoyl chloride hydrochloride (0.300 g, 1.685 mmol) and 3-pyridylcarbinol (0.184 g, 0.16 mL, 1.685 mmol) in dry dichloromethane (DCM) (30 mL) maintained at 0–5 °C. The mixture was stirred at room temperature for 24 h under nitrogen atmosphere. In order to neutralize the acid, NaHCO$_3$ solution (10% w/v) was added slowly to the mixture until the evolution of CO$_2$ has ceased. The organic layer was washed with distilled water, separated and dried over anhydrous sodium sulfate. Purification of the crude product by column chromatography

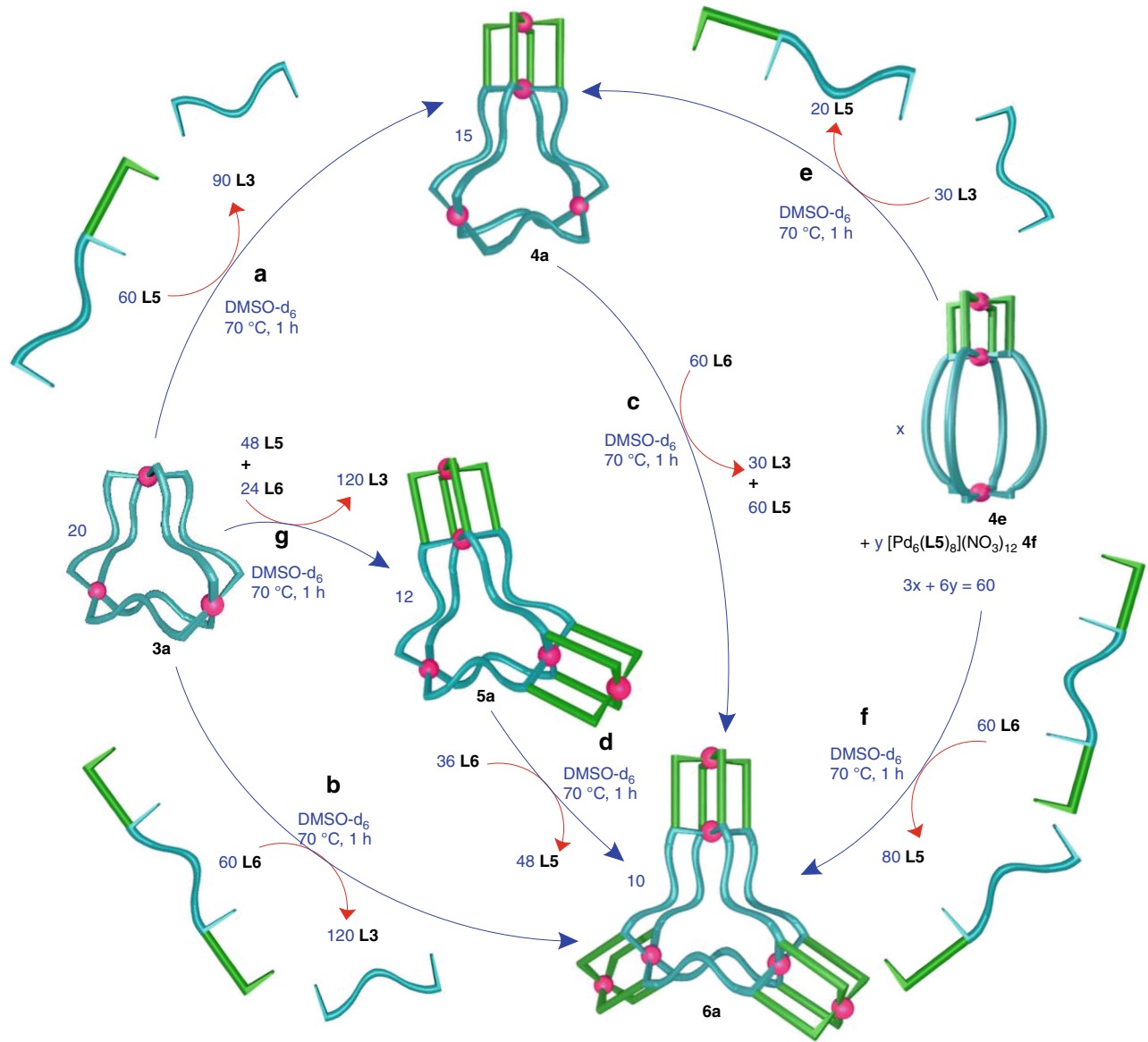

**Fig. 6 Ligand-displacement-induced cage-to-cage transformations.** Initial-cage/ligand-input/final-cage/displaced-ligand **a** system **3a/L5/4a/L3**; **b** system **3a/L6/6a/L3**; **c** system **4a/L6/6a/L3**&**L5**; **d** system **5a/L6/6a/L5**; **e** system **4e/L3/4a/L5**; **f** system **4e/L6/6a/L5**; **g** system **3a/L5**&**L6/5a/L3** (stoichiometry are provided in the figure. The complex **4e** coexists with **4f**).

(ethyl acetate:hexane, 4:6) yielded the ligand **L2** as a colorless liquid (0.288 g, yield 80%) (see Supplementary Figs. 1–6).

TLC (ethyl acetate:hexane, 40:60 v/v): $R_f = 0.4$; [1]H NMR (500 MHz, DMSO-$d_6$, room temperature): $\delta$ 9.12 (d, $J = 1.3$ Hz, 1H, $H_{f2}$), 8.82 (dd, $J_1 = 4.8$ Hz, $J_2 = 1.6$ Hz, 1H, $H_{i2}$), 8.72 (d, $J = 1.4$ Hz, 1H, $H_{a2}$), 8.57 (m, 1H, $H_{b2}$), 8.32 (m, 1H, $H_{g2}$), 7.93 (d, $J = 7.8$ Hz, 1H, $H_{d2}$), 7.57 (m, 1H, $H_{h2}$), 7.43 (m, 1H, $H_{c2}$), 5.43 (s, 2H, $H_{e2}$); [13]C NMR (125 MHz, DMSO-$d_6$, room temperature): $\delta$ 164.6, 153.8, 150.1, 149.5, 149.4, 137.0, 136.1, 131.5, 125.4, 124.0, 123.7, 64.3; HRMS (ESI, CH$_2$Cl$_2$/CH$_3$OH): $m/z$ Calcd. for C$_{12}$H$_{10}$N$_2$O$_2$: 214.2200, found 215.0820 [M + H]$^+$.

Triethylamine (1.02 mL, 7.246 mmol) was added in a dropwise manner to a stirred suspension of nicotinoyl chloride hydrochloride (1.290 g, 7.246 mmol) and resorcinol (0.400 g, 3.633 mmol) in dry DCM (50 mL) maintained at 0–5 °C. The mixture was stirred at room temperature for 24 h under nitrogen atmosphere. In order to neutralize the acid, NaHCO$_3$ solution (10% *w/v*) was added slowly to the mixture until the evolution of CO$_2$ has ceased. The organic layer was washed with distilled water, separated and dried over anhydrous sodium sulfate. Complete evaporation of the solvent yielded the ligand **L3** as an off-white solid (0.872 g, yield 75%) (see Supplementary Figs. 18–22).

Melting point: 235 °C; [1]H NMR (500 MHz, DMSO-$d_6$, room temperature): $\delta$ 9.28 (m, 2H, $H_{a3}$), 8.91 (m, 2H, $H_{b3}$), 8.48 (m, 2H, $H_{d3}$), 7.66 (m, 2H, $H_{c3}$), 7.60 (t, $J = 8.3$ Hz, 1H, $H_{f3}$), 7.43 (t, $J = 2.3$ Hz, 1H, $H_{g3}$), 7.34 (m, 2H, $H_{e3}$); [13]C NMR (125 MHz, DMSO-$d_6$, room temperature): $\delta$ 163.4, 154.3, 150.8, 150.5, 137.5, 130.2,

125.0, 124.1, 119.9, 116.2; HRMS (ESI, CH$_2$Cl$_2$/CH$_3$OH): $m/z$ Calcd. for C$_{18}$H$_{12}$N$_2$O$_4$: 320.2989, Found 321.0680 [M + H]$^+$.

The ligand, **L1** (0.500 g, 1.431 mmol) was dissolved in 35 mL of THF:H$_2$O (1:1) to obtain a clear solution. To this 1 mL of 2 N KOH was added and stirred at room temperature for 2 min, followed by addition of 0.5 mL of 4 N HCl. The reaction mixture was dried under vacuo yielding a solid (0.358 g, 97%), which was further purified by column chromatography using DCM:MeOH (3:7) to obtain the ligand **L4**. (see Supplementary Figs. 33–38).

Melting point: 223 °C; TLC (DCM:MeOH, 97:3 v/v): $R_f = 0.3$; [1]H NMR (500 MHz, DMSO-$d_6$, room temperature): $\delta$ 9.31 (s, 1H, $H_f/H_h$), 9.28 (s, 1H, $H_f/H_h$), 8.75 (m, 1H, $H_a$), 8.65 (m, 1H, $H_g$), 8.60 (m, 1H, $H_b$), 7.98 (m, 1H, $H_d$), 7.48 (m, 1H, $H_c$), 5.46 (s, 2H, $H_e$); [13]C NMR (125 MHz, DMSO-$d_6$, room temperature): $\delta$ 165.3, 163.9, 154.0, 153.4, 149.4, 149.3, 137.3, 136.4, 131.4, 126.8, 125.5, 123.8, 64.7; HRMS (ESI, CH$_2$Cl$_2$/CH$_3$OH): $m/z$ Calcd. for C$_{13}$H$_{10}$N$_2$O$_4$: 258.2295, Found 259.0722 [M + H]$^+$.

To a suspension of nicotinic acid (0.500 g, 4.065 mmol) and resorcinol (0.112 g, 1.018 mmol) in 50 mL dry DCM maintained at 0–5 °C, dimethylaminopyridine (DMAP) (0.062 g, 0.507 mmol) was added followed by *N*-ethyl-*N'*-(3-dimethylaminopropyl)carbodiimide hydrochloride (EDC.HCl) (0.195 g, 1.017 mmol). The reaction mixture was stirred for 12 h under nitrogen atmosphere. In order to neutralize the acid, NaHCO$_3$ solution (10% *w/v*) was added slowly to the mixture until the evolution of CO$_2$ has ceased. The organic layer was washed with distilled water, separated and dried over anhydrous sodium

sulfate. On standing the product precipitated out. It was isolated by filtration and dried under vacuo to obtain the ligand **L4′** as a pale pink powder. (0.150 g, 68%) (see Supplementary Figs. 39–43).

$^1$H NMR (500 MHz, DMSO-$d_6$, room temperature): δ 9.80 (d, $J$ = 7.5 Hz, 1H, –OH), 9.24 (m, 1H, H$_a$), 8.89 (m, 1H, H$_b$), 8.44 (m, 1H, H$_d$), 7.64 (m, 1H, H$_c$), 7.25 (m, 1H), 6.71 (dd, $J_1$ = 8.2 Hz, $J_2$ = 2.0 Hz, 3H); $^{13}$C NMR (125 MHz, DMSO-$d_6$, room temperature): δ 163.5, 158.4, 154.2, 151.2 150.5, 137.5, 130.0, 125.2, 124.1, 113.3, 112.2, 109.1.

To a suspension of 5-((pyridine-3-ylmethoxy)carbonyl)nicotinic acid, **L4** (0.250 g, 0.968 mmol) and 3-hydroxyphenyl nicotinate, **L4′** (0.208 g, 0.968 mmol) in 15 mL dry dimethylformamide (DMF) maintained at 0–5 °C, DMAP (0.059 g, 0.484 mmol) was added followed by addition of EDC·HCl (0.186 g, 0.968 mmol). The reaction mixture was stirred for 24 h under nitrogen atmosphere. Addition of water to the reaction mixture resulted in the precipitation of product. The product was isolated by filtration and dried under vacuo to obtain the ligand **L5** as an off-white solid powder (0.287 g, 65%). (see Supplementary Figs. 44–49).

Melting point: 238 °C; $^1$H NMR (500 MHz, DMSO-$d_6$, room temperature): δ 9.48 (d, $J$ = 2.1 Hz, 1H, H$_{f5}$/H$_{h5}$), 9.41 (d, $J$ = 2.0 Hz, 1H, H$_{f5}$/H$_{h5}$), 9.27 (d, $J$ = 1.4 Hz, 1H, H$_{m5}$), 8.91 (m, 1H, H$_{p5}$), 8.83 (s, 1H, H$_{g5}$), 8.75 (s, 1H, H$_{a5}$), 8.58 (m, 1H, H$_{b5}$), 8.49 (m, 1H, H$_{n5}$), 7.88 (m, 1H, H$_{d5}$), 7.67 (m, 1H, H$_{o5}$), 7.61 (t, $J$ = 8.2 Hz, 1H), 7.45 (m, 2H), 7.36 (m, 2H), 5.49 (s, 2H, H$_{e5}$); $^{13}$C NMR (125 MHz, DMSO-$d_6$, room temperature): δ 163.7, 163.4, 162.6, 154.3, 154.2, 150.8, 150.7, 150.6, 149.5, 149.5, 137.7, 137.5, 136.2, 131.3, 130.2, 125.7, 125.3, 124.9, 124.1, 123.7, 120.0, 119.8, 116.1, 64.8; HRMS (ESI, DCM:MeOH): $m/z$ Calcd. for C$_{25}$H$_{17}$N$_3$O$_6$: 455.4190, Found 456.1196 [M + H]$^+$.

To a suspension of 5-((pyridine-3-ylmethoxy)carbonyl)nicotinic acid, **L4** (0.469 g, 1.816 mmol) and resorcinol (0.100 g, 0.908 mmol) in 15 mL dry DMF maintained at 0–5 °C, DMAP (0.055 g, 0.4541 mmol) was added followed by EDC·HCl (0.348 g, 1.816 mmol). The reaction mixture was stirred for 24 h under nitrogen atmosphere. Addition of water to the reaction mixture resulted in the precipitation of product. The product was isolated by filtration and dried under vacuo to obtain the ligand **L6** as an off-white solid. (see Supplementary Figs. 63–68).

Melting point: 251 °C; $^1$H NMR (500 MHz, DMSO-$d_6$, room temperature): δ 9.49 (s, 2H, H$_{f6}$/ H$_{h6}$), 9.42 (s, 2H, H$_{h6}$/H$_{f6}$), 8.83 (s, 2H, H$_{g6}$), 8.75 (s, 2H, H$_{a6}$), 8.58 (m, 2H, H$_{b6}$), 7.97 (m, 2H, H$_{d6}$), 7.62 (t, $J$ = 8.3 Hz, 3H, H$_{j6}$), 7.46 (m, 2H, H$_{c6}$, H$_{k6}$), 7.37 (m, 2H, H$_{i6}$), 5.49 (s, 4H, H$_{e6}$); $^{13}$C NMR (125 MHz, DMSO-$d_6$, room temperature): δ 163.7, 162.6, 154.2, 150.6, 149.6, 149.5, 137.7, 136.2, 131.3, 130.3, 125.7, 125.3, 123.7, 123.7, 120.0, 116.1, 64.8; HRMS (ESI, DCM:MeOH): $m/z$ Calcd. for C$_{32}$H$_{22}$N$_4$O$_8$: 590.5391, Found 591.1501 [M + H]$^+$.

### Synthesis and characterization of the complexes.

The ligand **L2** (12.85 mg, 0.059 mmol) was added to a solution of Pd(NO$_3$)$_2$ (6.91 mg, 0.029 mmol) in 3 mL of DMSO. The reaction mixture was stirred at room temperature for 10 min subsequent, addition of 10 mL of ethyl acetate to the reaction mixture precipitated a white solid, which was separated by centrifugation. The solid was washed with 2 × 2 mL of acetone and dried under vacuum to obtain the complex [NO$_3$ ⊂ **Pd$_2$(L2)$_4$](NO$_3$)$_3$, 2a** as a mixture of diastereomers (16.80 mg, isolated yield 85%) (see Supplementary Figs. 7–13).

Melting point: 232 °C (decomposed); $^1$H NMR (500 MHz, DMSO-$d_6$, room temperature): δ 9.12 (1H, H$_{f2}$), 8.82 (1H, H$_{i2}$), 8.72 (1H, H$_{a2}$), 8.57 (1H, H$_{b2}$), 8.33 (1H, H$_{g2}$), 7.94 (1H, H$_{d2}$), 7.57 (1H, H$_{h2}$), 7.44 (1H, H$_{c2}$), 5.43 (1H, H$_{e2}$) [multiplicity has not been given as it is a mixture of isomers]; $^{13}$C NMR (125 MHz, DMSO-$d_6$, room temperature): δ 162.3, 155.0, 153.2, 153.1, 150.6, 149.3, 149.2, 141.9, 139.3, 139.23, 134.9, 128.7, 128.6, 127.6, 126.4, 118.0, 64.7. DOSY NMR (500 MHz, DMSO-$d_6$, 298 K): $D$ = 1.12 × 10$^{-10}$ m$^2$ s$^{-1}$; HRMS (ESI, DMSO): $m/z$ Calcd. for [**2a**–1•NO$_3$]$^{1+}$ 1256.0705, found 1256.0688; Calcd. for [**2a**–2•NO$_3$]$^{2+}$ 597.0416, found 597.0410; Calcd. for [**2a**–3•NO$_3$]$^{3+}$ 377.3651, found 377.3636.

Addition of appropriate TBAX to the complex **2a**, afforded the complexes [X ⊂ **Pd$_2$(L2)$_4$](NO$_3$)$_3$, 2b–2d**. (see Supplementary Methods and Supplementary Figs. 14, 15).

A solution of Pd(BF$_4$)$_2$ was prepared in 0.4 mL of DMSO-$d_6$ by stirring a mixture of PdI$_2$ (1.80 mg, 0.005 mmol) and AgBF$_4$ (1.95 mg, 0.010 mmol) at 90 °C for 30 min The precipitated AgI was separated by centrifugation. The ligand **L2** (2.14 mg, 0.010 mmol) was added to the supernatant and the reaction mixture was stirred at 70 °C for 2 h. To this solution tetra-$n$-butylammonium nitrate (0.76 mg, 0.0026 mmol) was added and heated for 5 min at 70 °C, resulting in the complex [NO$_3$ ⊂ **Pd$_2$(L2)$_4$](BF$_4$)$_4$, 2a′**. The $^1$H NMR spectrum of the complex **2a′** is closely comparable with that of complex **2a**.

HRMS (ESI, DMSO): $m/z$ Calcd. for [**2a′**–1•BF$_4$]$^{1+}$ 1305.1025, found 1305.0914; Calcd. for [**2a′**–2•BF$_4$]$^{2+}$ 609.5496, found 609.5459; Calcd. for [**2a′**–3•BF$_4$]$^{3+}$ 377.3651, found 377.3636.

Complexes [X ⊂ **Pd$_2$(L2)$_4$](NO$_3$)$_3$, 2b′–2d′** were prepared in a similar way. (see Supplementary Methods and Supplementary Figs. 16, 17).

The ligand **L3** (19.21 mg, 0.059 mmol) was added to a solution of Pd(NO$_3$)$_2$ (6.91 mg, 0.029 mmol) in 3 mL of DMSO. The reaction mixture was stirred at room temperature for 10 min. Subsequent, slow diffusion of toluene vapor into reaction mixture precipitated a crystalline solid, which was separated by filtration. The solid was dried under vacuum to yield the complex [**Pd$_3$(L3)$_6$](NO$_3$)$_6$, 3a** (22.72 mg, isolated yield 58%) (see Supplementary Figs. 23–27).

Melting point: 258 °C (decomposed); $^1$H NMR (500 MHz, DMSO-$d_6$, room temperature): δ 10.24 (bs, 2H, H$_{a3}$), 9.68 (bs, 2H, H$_{b3}$), 8.65 (bs, 2H, H$_{d3}$), 7.98 (m, 2H, H$_{c3}$), 7.67 (t, $J$ = 8.15 Hz, 1H, H$_{f3}$), 7.37 (m, 2.3 Hz, 3H, H$_{e3}$/H$_{g3}$); $^{13}$C NMR (125 MHz, DMSO-$d_6$, room temperature): δ 161.7, 154.9, 152.6, 150.7, 141.6, 130.7, 128.6, 127.7, 120.6, 116.8; DOSY NMR (500 MHz, DMSO-$d_6$, 298 K): $D$ = 9.54 × 10$^{-11}$ m$^2$ s$^{-1}$; HRMS (ESI, DMSO): $m/z$ Calcd. for [**3a**–2•NO$_3$]$^{2+}$ 1244.0733, found 1244.0722; Calcd. for [**3a**–5•NO$_3$]$^{5+}$ 460.4369, found 460.4373.

A solution of Pd(BF$_4$)$_2$ was prepared in 0.5 mL of DMSO-$d_6$ by stirring a mixture of PdI$_2$ (1.80 mg, 0.005 mmol) and AgBF$_4$ (1.95 mg, 0.010 mmol) at 90 °C for 30 min The precipitated AgI was separated by centrifugation. The ligand **L3** (3.20 mg, 0.010 mmol) was added to the supernatant and the reaction mixture was stirred at room temperature for 10 min to afford the complex [**Pd$_3$(L3)$_6$](BF$_4$)$_6$, 3b**. The $^1$H NMR spectrum of the complex **3b** is closely comparable with that of complex **3a** except for H$_{g3}$, 7.30 (s, 1H, H$_{g3}$). (see Supplementary Fig. 29).

HRMS (ESI, DMSO): $m/z$ Calcd. for [**3b**–3•BF$_4$]$^{3+}$ 833.7355, found 833.7359; Calcd. for [**3b**–4•BF$_4$]$^{4+}$ 603.5511, found 603.5510; Calcd. for [**3b**–5•BF$_4$]$^{5+}$ 465.4402, found 465.4403.

Complexes [**Pd$_3$(L3)$_6$](X)$_6$, 3c–3f** were prepared in a similar way. (see Supplementary Methods and Supplementary Fig. 28).

The ligands **L3** (8.00 mg, 0.025 mmol) and **L5** (22.77 mg, 0.050 mmol) were added to a solution of Pd(NO$_3$)$_2$ (11.52 mg, 0.050 mmol) in 5 mL of DMSO. The reaction mixture was stirred at 70 °C for 1 h to obtain a clear solution. Subsequent, slow diffusion of toluene vapor to the resulting solution precipitated a crystalline solid, which was separated by filtration. The solid was dried under vacuum to obtain the complex [NO$_3$ ⊂ **Pd$_4$(L3)$_2$(L5)$_4$](NO$_3$)$_7$, 4a** (18.98 mg, isolated yield 45%). (see Supplementary Figs. 50–58).

Melting point: 260 °C (decomposed); $^1$H NMR (500 MHz, DMSO-$d_6$, room temperature): δ 11.02 (s, 4H, H$_{f5}$), 10.65 (s, 4H, H$_{h5}$), 10.25 (d, $J$ = 3.5 Hz, 8H, H$_{m5}$, H$_{a3}$), 10.05 (s, 4H, H$_{a5}$), 9.66 (d, $J$ = 4.2 Hz, 8H, H$_{p5}$, H$_{b3}$), 9.35 (d, $J$ = 4.5 Hz, 4H, H$_{b5}$), 8.92 (s, 4H, H$_{g5}$), 8.66 (d, $J$ = 6.5 Hz, 8H, H$_{n5}$, H$_{d3}$), 8.16 (m, 4H, H$_{d5}$), 8.00–7.97 (m, 8H, H$_{o5}$, H$_{c3}$), 7.82 (m, 4H, H$_{c5}$), 7.70–7.66 (m, 4H), 7.41–7.35 (m, 16H), 5.56 (m, 8H, H$_{e5}$); $^{13}$C NMR (125 MHz, DMSO-$d_6$, room temperature): δ 161.9, 161.8, 161.3, 156.6, 155.0, 152.6, 150.8, 150.8, 149.4, 141.6, 139.5, 134.8, 130.8, 129.1, 129.0, 128.7, 128.6, 127.8, 126.7, 120.9, 120.7, 117.0, 65.3; DOSY NMR (500 MHz, DMSO-$d_6$, 298 K): $D$ = 8.51 × 10$^{-11}$ m$^2$ s$^{-1}$; HRMS (ESI, DMSO): $m/z$ Calcd. for [**4a**–3•NO$_3$]$^{3+}$ 1065.7228, found 1065.7184; Calcd. for [**4a**–6•NO$_3$]$^{6+}$ 501.8678, found 501.8661.

Addition of appropriate TBAX to the complex **4a**, afforded the complexes X ⊂ **Pd$_4$(L1)$_2$(L2)$_4$](NO$_3$)$_7$, 4b–4d**. (see Supplementary Methods and Supplementary Figs. 59, 60, 61).

A solution of Pd(BF$_4$)$_2$ was prepared in 0.4 mL of DMSO-$d_6$ by stirring a mixture of PdI$_2$ (1.80 mg, 0.005 mmol) and AgBF$_4$ (1.95 mg, 0.010 mmol) at 90 °C for 30 min The precipitated AgI was separated by centrifugation. The ligands **L3** (0.80 mg, 0.003 mmol) and **L5** (2.28 mg, 0.005 mmol) were added the supernatant and heated at 70 °C for 2 h, yielding an oligomer. To the solution containing the oligomer, tetra-$n$-butylammonium nitrate (0.38 mg, 0.0012 mmol) in 0.1 mL of DMSO-$d_6$ was added and heated at 70 °C for 1 h resulting in the formation of complex [NO$_3$ ⊂ **Pd$_4$(L3)$_2$(L5)$_4$](BF$_4$)$_7$, 4a′**. The $^1$H NMR spectrum of the complex **4a′** is closely comparable with the data of complex **4a**.

Complexes [X ⊂ **Pd$_4$(L3)$_2$(L5)$_4$](BF$_4$)$_7$, 4b′–4d′** were prepared in a similar way. (see Supplementary Methods and Supplementary Fig. 62).

The ligands **L5** (18.22 mg, 0.040 mmol) and **L6** (11.81 mg, 0.020 mmol) were added to a solution of Pd(NO$_3$)$_2$ (11.52 mg, 0.050 mmol) in 5 mL of DMSO. The reaction mixture was stirred at 70 °C temperature for 2 h. Subsequently, slow diffusion of toluene vapor to the reaction mixture precipitated a crystalline solid, which was separated by filtration. The solid was dried under vacuum to obtain the complex [(NO$_3$)$_2$ ⊂ **Pd$_5$(L5)$_4$(L6)$_2$](NO$_3$)$_8$, 5a** (15.76 mg, isolated yield 38%). (see Supplementary Figs. 69–77).

Melting point: 272 °C (decomposed); $^1$H NMR (500 MHz, DMSO-$d_6$, room temperature): δ 11.04 (s, 8H, H$_{f5}$, H$_{f6}$), 10.67 (s, 8H, H$_{h5}$, H$_{h6}$), 10.26 (d, 4H, H$_{m5}$), 10.06 (s, 8H, H$_{a5}$, H$_{a6}$), 9.68 (s, 4H, H$_{p5}$), 9.36 (d, 8H, H$_{b5}$, H$_{b6}$), 8.94 (s, 8H, H$_{g5}$, H$_{g6}$), 8.67 (d, $J$ = 8.2 Hz, 4H, H$_{n5}$), 8.18 (bs, 8H, H$_{d5}$, H$_{d6}$), 8.01–7.98 (m, 4H, H$_{o5}$), 7.85–7.82 (m, 8H, H$_{c5}$, H$_{c6}$), 7.75–7.71 (m, 4H), 7.48–7.41 (m, 16H), 5.64–5.53 (m, 16H, H$_{e5}$, H$_{e5}$); $^{13}$C NMR (125 MHz, DMSO-$d_6$, room temperature): δ 161.8, 161.3, 156.5, 150.8, 149.3, 141.5, 139.5, 134.8, 130.8, 128.9, 128.7, 126.6, 120.9, 116.8, 65.2; DOSY NMR (500 MHz, DMSO-$d_6$, 298 K): $D$ = 5.75 × 10$^{-11}$ m$^2$ s$^{-1}$; HRMS (ESI, DMSO): $m/z$ Calcd. for [**5a**–3•NO$_3$]$^{3+}$ 1322.7259, found 1322.7248; Calcd. for [**5a**–4•NO$_3$]$^{4+}$ 976.5475, found 976.5474.

Addition of appropriate TBAX to the complex **5a**, provided the complexes [(X)$_2$ ⊂ **Pd$_5$(L5)$_4$(L6)$_2$](NO$_3$)$_8$, 5b–5d**. (see Supplementary Methods and Supplementary Figs. 78–81).

A solution of Pd(BF$_4$)$_2$ was prepared in 0.4 mL of DMSO-$d_6$ by stirring a mixture of PdI$_2$ (1.80 mg, 0.005 mmol) and AgBF$_4$ (1.95 mg, 0.010 mmol) at 90 °C for 30 min The precipitated AgI was separated by centrifugation. The ligands **L5** (1.82 mg, 0.004 mmol) and **L6** (1.18 mg, 0.002 mmol) were added to the supernatant and heated at 70 °C for 2 h to obtain the oligomer. To the solution containing the oligomer, a solution of tetra-$n$-butylammonium nitrate (0.61 mg, 0.002 mmol) in 0.1 mL of DMSO-$d_6$ was added and heated at 70 °C for 1 h resulting in the complex [(NO$_3$)$_2$ ⊂ **Pd$_5$(L5)$_2$(L6)$_4$](NO$_3$)$_8$, 5a′**. The $^1$H NMR spectrum of the complex **5a′** is closely comparable with the data of complex **5a**.

Complexes $[(X)_2 \subset Pd_5(L5)_2(L6)_4](NO_3)_8$, **5a′–5d′** were prepared in a similar way. (see Supplementary Methods and Supplementary Fig. 82).

The ligand **L6** (29.53 mg, 0.050 mmol) was added to a solution of $Pd(NO_3)_2$ (11.52 mg, 0.050 mmol) in 5 mL of DMSO. The reaction mixture was stirred at 70 °C for 20 min to obtain a clear solution. Subsequently, addition of 15 mL of ethyl acetate to the reaction mixture precipitated a white solid, which was separated by centrifugation. The solid was washed with 3 × 3 mL of acetone and dried under vacuum to obtain the complex $[(NO_3) \subset Pd_6(L6)_6](NO_3)_9$, **6a** (36.98 mg, isolated yield 90%). (see Supplementary Figs. 83–88).

Melting point: 288 °C (decomposed); $^1H$ NMR: 11.04 (s, 12H, $H_{f6}$), 10.68 (s, 12H, $H_{h6}$), 10.06 (s, 12H, $H_{a6}$), 9.37 (d, J = 8.2 Hz, 12H, $H_{b6}$), 8.94 (s, 12H, $H_{g6}$), 8.19 (d, J = 7.2 Hz, 12H, $H_{d6}$), 7.80 (s, 12H, $H_{c6}$), 7.75 (bs, 6H), 7.48 (bs, 18H), 5.58 (bs, 24H, $H_{e5}$); HRMS (ESI, DMSO): $m/z$ Calcd. for $[6a–5·NO_3]^{5+}$ 923.0424, found 923.0411.

Addition of appropriate TBAX to the complex **6a**, resulted in the complexes $[(X)_3 \subset Pd_6(L6)_6](NO_3)_9$, **6b–6d**. (see Supplementary Methods and Supplementary Figs. 89–93).

A solution of $Pd(BF_4)_2$ in 0.4 mL of DMSO-$d_6$ was prepared by stirring a mixture of $PdI_2$ (1.80 mg, 0.005 mmol) and $AgBF_4$ (1.95 mg, 0.010 mmol) at 90 °C for 30 min The precipitated AgI was separated by centrifugation. The ligand **L6** (2.95 mg, 0.005 mmol) was added to the supernatant and heated at 70 °C for 2 h to obtain the oligomer. To the solution containing the oligomer, a solution of tetra-$n$-butylammonium nitrate (0.76 mg, 0.0025 mmol) was added and heated at 70 °C for 20 min resulting the complex $[(NO_3) \subset Pd_6(L6)_6](BF_4)_9$, **6a′**. The $^1H$ NMR spectrum of the complex **6a′** is closely comparable with the data of complex **6a**.

Complexes $[(X)_3 \subset Pd_6(L6)_6](BF_4)_9$, **6b′–6d′** were prepared in a similar way. (see Supplementary Methods and Supplementary Fig. 94).

## Data availability

The authors declare that the data supporting the findings of this study are available within the Supplementary Information files and from the corresponding author upon reasonable request. Comparison of chemical shift values 4a–4d, 5a–5d, and 6a–6d are given in Supplementary Tables 2–4. The X-ray crystallographic coordinates for structures reported in this study have been deposited at the Cambridge Crystallographic Data Center (CCDC), under deposition numbers 1941617-1941622. These data can be obtained free of charge from The Cambridge Crystallographic Data Center via www.ccdc.cam.ac.uk/data_request/cif. Crystallographic data are given in Supplementary Tables 5 and 6.

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

## Acknowledgements

D.K.C. thanks the Science and Engineering Research Board (SERB), Department of Science and Technology, Government of India (project no. EMR/2017/002262), for financial support. D.K.C. acknowledges financial support provided by IIT Madras under the Mid-Career Institute Research and Development Award (IRDA-2019). S.S. thanks CSIR for a fellowship. S.S. thanks R.D. Chakravarthy for helpful discussion during the synthesis of ligands. S.K. thanks IIT Madras for an Institute Postdoctoral Fellowship. We thank SAIF, IIT Madras for single crystal XRD facility and Department of Chemistry, IIT Madras for DST-FIST funded ESI-MS facility. We thank Agilent Technologies India Pvt. Ltd. for valuable assistance in ESI-MS data collection.

## Author contributions

D.K.C. and S.S. designed the work, carried out research and analyzed data. S.S. was involved in single crystal growth of all the cage molecules and contributed in composing the manuscript. S.K. handled the crystals and solved the crystal structures. D.K.C. is the principal investigator of the project and wrote the manuscript. All the authors discussed the work and edited the manuscript.

## Competing interests

The authors declare no competing interests.
