## [Peer Review File · Nature Communications]

Reviewer #1 (Remarks to the Author):

Review for NCOMMS-19-25206-T

Self-assembled conjoined-cages

The manuscript by Chand and co-worker is simply superb, while the development of self-assembled cages is now very common it has so far been dominated by access only to single cavity symmetric cage systems. This limits the potential applications of the cage materials. Because of this many groups around the world, including my own team, are attempting to generate multicavity cages to widen the potential applications of molecules cages.

Herein Chand and his team generate a series of amazing multicavity cages using some new polypyridyl ligands and Pd(II) ions. This work is spectacular, the ligand synthesis alone is extremely good but cages assemblies are amazing and all the materials are well characterised using NMR and MS.

A series multi-cavity cages of featuring [Pd₄(La)₂(Lb)₄], [Pd₅(Lb)₄(Lc)₂] and [Pd₆(Lc)₆] formulations, respectively were synthesised and impressively the molecular architectures established indisputably from their single crystal X-ray structures.

While limited in scope due the size of the cage cavities the authors also show that the different cavities can bind guest molecules. The Pd₂L₄ cavities interacting with anions while the larger “central” cavity between the Pd₂L₄ cavities able to bind solvents such as DMSO molecules.

This is a brilliant piece of work even better than the molecular star previously reported by this group and should be published in Nature Commun.

Minor revisions

Some of the language in place could be improved/sharpened

Eg. “...electively bind NO₃, F, Cl or Br anion while the central bigger cavity accommodated up to four DMSO molecules” change to “...electively bind NO₃⁻, F⁻, Cl⁻ or Br⁻ anions while the central bigger cavity accommodated up to four DMSO molecules”.

The authors state “The Crowley group introduced an additional donor site in the ligand, creating a tetrakismonodentate ligand that allowed the formation of a [Pd₄L₄] type complex with three identical 3D cavities arranged in a linear fashion.²⁹”

This statement is not quite correct I think the Crowley system has two different types of cavities (three cavities in total) arranged in a linear fashion

The author call the isomers of cage 2a “spatial isomers” while this is fine I think they more appropriately called diastereomers

Reviewer #2 (Remarks to the Author):

This is a very interesting paper by Chand and co-workers that describes the self-assembly of a series of oligopyridyl ligands in combination with Pd ions. Quite interestingly, the authors find that they can create a series of conjoined cages (2, 3, or 4 cages) whereby the sides of adjacent cages are fused together in a non-linear fashion. Beyond the beauty of the self-assemblies is the very interesting transformations of them. For example, they can undergo integrative or narcissistic self-sorting when multiple components are mixed at the same time. Furthermore, the high fidelity cage fusion processes are really quite interesting. Overall, the results are very high quality, the structures are striking so I am very happy to recommend publication after revisions.

1) I found the text of the manuscript to be quite repetitive in several places (e.g. description of anion exchange) and therefore rather tedious to read. The role of the counter anions is never really clearly delineated so this is confusing to the reader as is the "*" nomenclature in the paper.

2) Conjoined cages are of course found in MOFs. I suggest the authors add a sentence or two and a reference to point this out.

3) The referencing of earlier work in the field is quite extensive (e.g. sentence #1). I suggest just referring to a couple authoritative reviews.

4) The language could benefit from a revision by a native speaker or the editorial office.

5) In Figure 6, the displacing ligands have poor quality in the pdf file. Please correct.

Reviewer #3 (Remarks to the Author):

The manuscript by Chand et al reports on design and structural (NMR, ESI-MS and SCXRD) studies of self-assembled coordination cages, which due to their multiple cavities, are called by the authors as conjoined-cages.

The ms presents an extension of the earlier work by the authors, but also many others using the same ideology (Lehn, Saalfrank, Sauvage, Fujita, Stang, Cotton, Raymond, Newkome, etc..). The design is based on the very well-known Pd(py)₄ (2+) coordination motif, where a square planar complex is formed. Depending how the pyridine ligand is designed, multitude of coordination complexes, either molecular or polymeric, can be envisaged.

Dr. Chand uses in his design simple pyridine carboxylic acid esters, which have auxiliary pyridine moieties attached. These ligands are semi-linear and have a lot of degrees of rotational freedom around the ester and O-CH-Pyridine bonds.

Despite the aesthetic beauty of the presented multicavity coordination cages, so that the larger cavity binds solvent molecules, while the smaller ones bind, they are not the only such systems. Prof. Luetzen (Univ. Bonn, Germany) published already in 2014 a multicavity coordination cage based on the same Pd(py)₄ (2+) coordination motif (Angew. Chem. Int. Ed. 2014, 53, 3739–3742.). The authors are encouraged to cite this and may other Luetzen work.

In addition to the omission of an important similar work, this ms has several discrepancies between the main text and SI.

Particularly alarming in the X-ray work is the preparation (XRD part of the SI) of the partially occupied chloride complexes using AgCl for the anion exchange from nitrate to chloride. AgCl is much more insoluble than AgNO₃, and this referee fails to understand how it could work. 35-25 % chlorine atom equals to 4 - 6 electrons and based on the uncertainty of the AgCl usage, it might be wishful thinking to believe partial chloride anion in the cavity.

The checkcif report are missing from the submission and there are some issues in the cif files, which should be at least commented or revised using the IUCR Output Validation Response Form, see <http://checkcif.iucr.org>

2c: Alert level B

PLAT084_ALERT_3_B High wR2 Value (i.e. > 0.25) 0.41 Report

PLAT342_ALERT_3_B Low Bond Precision on C-C Bonds 0.03 Ang.

3a: Alert level A

PLAT029_ALERT_3_A _diffrn_measured_fraction_theta_full value Low . 0.893 Why?

4a: Alert level A

PLAT213_ALERT_2_A Atom C88 has ADP max/min Ratio 5.1 prolat

PLAT910_ALERT_3_A Missing # of FCF Reflection(s) Below Theta(Min). 53 Note

4a': Alert level B

PLAT342_ALERT_3_B Low Bond Precision on C-C Bonds 0.02539 Ang.

5c: Alert level A

THETM01_ALERT_3_A The value of $\sin(\theta_{\max})/\lambda$ is less than 0.550

Calculated $\sin(\theta_{\max})/\lambda = 0.5056$

PLAT910_ALERT_3_A Missing # of FCF Reflection(s) Below Theta(Min). 70 Note

Alert level B

PLAT220_ALERT_2_B Non-Solvent Resd 1 C Ueq(max)/Ueq(min) Range 6.1 Ratio

PLAT241_ALERT_2_B High 'MainMol' Ueq as Compared to Neighbors of C6 Check

PLAT342_ALERT_3_B Low Bond Precision on C-C Bonds 0.02025 Ang.

6c: Alert level B

PLAT241_ALERT_2_B High 'MainMol' Ueq as Compared to Neighbors of C40 Check

PLAT910_ALERT_3_B Missing # of FCF Reflection(s) Below Theta(Min). 48 Note

Based on the above thgis referee can't support the publication of this work in Nature Communications.
After careful revisions a proper journal could be Chemical Science (or Inorganic Chemistry).

Response to reviewers' comments

Response to Reviewer #1

The manuscript by Chand and co-worker is simply superb, while the development of self-assembled cages is now very common it has so far been dominated by access only to single cavity symmetric cage systems. This limits the potential applications of the cage materials. Because of this many groups around the world, including my own team, are attempting to generate multicavity cages to widen the potential applications of molecules cages. Herein Chand and his team generate a series of amazing multicavity cages using some new polypyridyl ligands and Pd(II) ions. This work is spectacular, the ligand synthesis alone is extremely good but cages assemblies are amazing and all the materials are well characterised using NMR and MS.

A series multi-cavity cages of featuring [Pd₄(La)₂(Lb)₄], [Pd₅(Lb)₄(Lc)₂] and [Pd₆(Lc)₆] formulations, respectively were synthesised and impressively the molecular architectures established indisputably from their single crystal X-ray structures.

While limited in scope due the size of the cage cavities the authors also show that the different cavities can bind guest molecules. The Pd₂L₄ cavities interacting with anions while the larger “central” cavity between the Pd₂L₄ cavities able to bind solvents such as DMSO molecules.

This is a brilliant piece of work even better than the molecular star previously reported by this group and should be published in Nature Commun.

We thank the Reviewer #1 for kindly appreciating our molecular design, synthesis, and structural determination. We are delighted to receive a strong recommendation from the reviewer and her/his support for the publication of our manuscript in Nature Commun.

Minor revisions

Query 1: Some of the language in place could be improved/sharpened

Eg. “...electively bind NO₃, F, Cl or Br anion while the central bigger cavity accommodated up to four DMSO molecules” change to “...electively bind NO₃⁻, F⁻, Cl⁻ or Br⁻ anions while the central bigger cavity accommodated up to four DMSO molecules”.

Response 1: All instances of “NO₃, F, Cl or Br” have been replaced with “NO₃⁻, F⁻, Cl⁻ or Br⁻” in the revised manuscript and the supplementary information.

Query 2: The authors state “The Crowley group introduced an additional donor site in the ligand, creating a tetrakismonodentate ligand that allowed the formation of a [Pd₄L₄] type complex with three identical 3D cavities arranged in a linear fashion.²⁹” This statement is not quite correct I think the Crowley system has two different types of cavities (three cavities in total) arranged in a linear fashion

Response 2: We agree with the reviewer. The description is modified suitably and also noted below

“The Crowley group introduced an additional donor site in the ligand design, creating a tetrakis-monodentate ligand that allowed the formation of a [Pd₄L₄] complex with three 3D-cavities arranged in a linear fashion.²³ The environment of central cavity, by virtue of its position, has to be different from a terminal cavity, however, there are subtle differences in the frameworks of the central versus terminal cavities in the design of Crowley.”

Query 3: The author call the isomers of cage 2a “spatial isomers” while this is fine I think they more appropriately called diastereomers

Response 3: According to the reviewer suggestion the word “spatial isomers” has been replaced with “diastereomers” to make the description more appropriate.

Response to Reviewer #2

This is a very interesting paper by Chand and co-workers that describes the self-assembly of a series of oligopyridyl ligands in combination with Pd ions. Quite interestingly, the authors find that they can create a series of conjoined cages (2, 3, or 4 cages) whereby the sides of adjacent cages are fused together in a non-linear fashion. Beyond the beauty of the self-assemblies is the very interesting transformations of them. For example, they can undergo integrative or narcissistic self-sorting when multiple components are mixed at the same time. Furthermore, the high fidelity cage fusion processes are really quite interesting. Overall, the results are very high quality, the structures are striking so I am very happy to recommend publication after revisions.

We thank the reviewer for giving encouraging feedback about the work described in the manuscript and a positive recommendation. We appreciate the minor criticisms. Our responses to the specific comments are given below.

Query 1: I found the text of the manuscript to be quite repetitive in several places (e.g. description of anion exchange) and therefore rather tedious to read. The role of the counter

anions is never really clearly delineated so this is confusing to the reader as is the "*" nomenclature in the paper.

Response 1 Part a: As suggested by the reviewer, the description of anion exchange reactions has been edited in the revised manuscript. The following consolidated paragraph regarding the role of anions in the formation of the cages has been included in the 'Discussion' section of the revised manuscript.

"This article demonstrated the construction of multi-3D-cavity coordination cages via decoration of a $[\text{Pd}_3\text{L}_6]$ core with one or more $[\text{Pd}_2\text{L}_4]$ units in a linear or lateral fashion, respectively. The metal component used for the preparation of the cages was $\text{Pd}(\text{NO}_3)_2$ and the cages formed (**2a**, **4a**, **5a** and **6a**) were found to encapsulate NO_3^- in their $[\text{Pd}_2\text{L}_4]$ moieties. The encapsulated NO_3^- could be replaced by halides like F^- , Cl^- or Br^- by using corresponding TBAX. Notably, AgCl could be also used as a source of Cl^- ion. In fact, Clever *et al* has employed sparingly soluble AgCl as a source of Cl^- that displaced bound BF_4^- ion from the cavity of certain coordination cages so that AgCl got consumed and more soluble AgBF_4 remained in solution.⁴⁷ We have also demonstrated the use of AgCl where the Cl^- ion displaced bound NO_3^- ion from the cavity of some coordination cages whereupon the more soluble AgNO_3 remained in solution.⁴⁸ The requirement of the encapsulated anions in the creation of these assemblies was realized when attempts toward synthesizing Pd_2L_4 , $[\text{Pd}_4(\text{L}^a)_2(\text{L}^b)_4]$, $[\text{Pd}_5(\text{L}^b)_4(\text{L}^c)_2]$ and $[\text{Pd}_6(\text{L}^c)_6]$ cages using $\text{Pd}(\text{BF}_4)_2$ failed and the experiments led to the formation of a mixture of unidentified products. Probably, the formation of the $[\text{Pd}_2\text{L}_4]$ entity was hindered, due to repulsion between the closely placed metal ions, in the absence of an appropriate anionic template. This hindrance in turn prevented the building of the targeted *conjoined*-cages. The BF_4^- ion was found to be unsuitable as a template here. The addition of TBAX (for $\text{X} = \text{NO}_3^-$, F^- , Cl^- and Br^- , respectively) to these mixtures yielded the desired cages. Hence, the formation of Pd_2L_4 , $[\text{Pd}_4(\text{L}^a)_2(\text{L}^b)_4]$, $[\text{Pd}_5(\text{L}^b)_4(\text{L}^c)_2]$ and $[\text{Pd}_6(\text{L}^c)_6]$ cages is feasible only when the smaller cavity is occupied by NO_3^- , F^- , Cl^- or Br^- , irrespective of the counter anion present outside the cavity/cavities. Although the formation of the larger cavity (i.e. $[\text{Pd}_3\text{L}_6]$ entity) is anion independent, this did not help in the formation of corresponding *conjoined*-cages (**4a**, **5a** and **6a**) since the formation of $[\text{Pd}_2\text{L}_4]$ entity is essential."

Response 1 Part b: As per the reviewers comment on the use of "*" in naming complexes, we agree that it could be confusing to readers. The nomenclature of the complexes were

slightly modified and the * has been removed from the nomenclature. The complex labelled as $[\text{Pd}_2(\text{L3})_4](\text{NO}_3)_4$, **2a*** is now **3g**, $[\text{Pd}_4(\text{L3})_8](\text{NO}_3)_8$, **4a*** is represented as **3h**, similarly the trinuclear complex formed by ligand **L5** is labelled as $[\text{NO}_3\text{C}\text{Pd}_3(\text{L5})_4](\text{NO}_3)_5$, **4e** and the hexanuclear complex $[(\text{NO}_3)_2\text{C}\text{Pd}_6(\text{L5})_8](\text{NO}_3)_{10}$ is notified as **4f**, that were earlier represented as **3a*** and **6a***. Accordingly the corresponding Cl^- encapsulated complexes $[\text{Cl}\text{C}\text{Pd}_3(\text{L5})_4](\text{NO}_3)_5$ and $[(\text{NO}_3)_2\text{C}\text{Pd}_6(\text{L5})_8](\text{NO}_3)_{10}$ were labelled as **4g** and **4h** respectively, which were denoted as **3c*** and **6c*** previously. The crystal structure **4ac** and **4ac'** are renamed as **4acI** and **4acII**.

Query 2: Conjoined cages are of course found in MOFs. I suggest the authors add a sentence or two and a reference to point this out.

Response 2: The following sentence drawing attention to the difference between the conjoined cages observed in 3D metal-organic frameworks and the conjoined cages in discrete cages described in this work has been added in the revised manuscript. The added sentences is noted below

“It is pertinent to note that a variety of 3D-metal–organic frameworks possessing an infinite number of *conjoined*-cages in their architectures are known.⁽⁷⁾”

Query 3: The referencing of earlier work in the field is quite extensive (e.g. sentence #1). I suggest just referring to a couple authoritative reviews.

Response 3: As per the reviewer’s suggestion the references has been modified. Instead of 15 references now we have 9 references. The references that are more closer to this work and the references that were used as cross reference in later part of the manuscript are retained.

Query 4: The language could benefit from a revision by a native speaker or the editorial office.

Response 4: We have improved the language in the revised manuscript and the supplementary information to the best of our abilities.

Query 5: In Figure 6, the displacing ligands have poor quality in the pdf file. Please correct.

Response 5: The clarity of the displacing ligand **L3** was poor in the figure 6, due to pdf file. To avoid that the old figure 6 has been replaced with the new figure 6, where the clarity of the displacing ligand **L3** has been improved.

Response to Reviewer #3

The manuscript by Chand et al reports on design and structural (NMR, ESI-MS and SCXRD) studies of self-assembled coordination cages, which due to their multiple cavities, are called by the authors as conjoined-cages.

The ms presents an extension of the earlier work by the authors, but also many others using the same ideology (Lehn, Saalfrank, Sauvage, Fujita, Stang, Cotton, Raymond, Newkome, etc.). The design is based on the very well-known Pd(py)₄ (2+) coordination motif, where a square planar complex is formed. Depending how the pyridine ligand is designed, multitude of coordination complexes, either molecular or polymeric, can be envisaged.

Dr. Chand uses in his design simple pyridine carboxylic acid esters, which have auxilliary pyridine moieties attached. These ligands are semi-linear and have a lot of degrees of rotational freedom around the ester and O-CH-Pyridine bonds.

We thank the reviewer for meticulously reviewing our manuscript. It is true that many aspects as pointed out by the reviewer #3 is well established. However, the objectives of the work is somewhat different. We demonstrate here the process of making conjoined-cages and how this design can be applicable for making new conjoined-cages. Our responses to specific comments of the reviewers are given below.

Query 1: Despite the aesthetic beauty of the presented multicavity coordination cages, so that the larger cavity binds solvent molecules, while the smaller ones bind, they are not the only such systems. Prof. Luetzen (Univ. Bonn, Germany) published already in 2014 a multicavity coordination cage based on the same Pd(py)₄ (2+) coordination motif (Angew. Chem. Int. Ed. 2014, 53, 3739 –3742.). The authors are encouraged to cite this and may other Luentzen work.

Response 1: We envisioned the self-assembled coordination cages with (i) “one 3D-cavity” and (ii) “more than one 3D-cavity” as two different class of compounds. It is correct to state that a coordination cage possessing “one 3D-cavity” is composed of more than one 2D-rings in its overall build. True for all the coordination cages. However, such a 3D-cavity may possess either one binding site in its entirety (usual) in a mono-topic fashion or more than one binding sites in a multi-topic fashion (not usual).

While the work of Luetzen, pointed out by the reviewer, describes a beautiful system possessing “one 3D-cavity” nevertheless, the authors demonstrated the multi-topic capability of the cage for binding multiple guests within the single discrete cavity. However, it will be a good idea to cite the work of Luetzen in the present manuscript and give credit to the ability of the cage for binding multiple variety of guests. We thank the reviewer for the criticism and the suggested work is now cited appropriately, also including another publication of Fujita and co-workers. The following sentence is included in the manuscript:

“In contrast to the multiple binding sites of the multi-3D-cavity cages, there exist single-3D-cavity systems capable of accommodating multiple variety of guests in site specific manner.^{28,29}”

Query 2: In addition to the omission of an important similar work, this ms has several discrepancies between the main text and SI.

Response 2: The text of the manuscript and the supplementary information has been revised extensively in accordance with journal guidelines. The discrepancies in the schematic representation of the cages has been resolved. We hope that the inconsistencies in the original files have been satisfactorily addressed in the revised versions.

Query 3: Particularly alarming in the X-ray work is the preparation (XRD part of the SI) of the partially occupied chloride complexes using AgCl for the anion exchange from nitrate to chloride. AgCl is much more insoluble than AgNO₃, and this referee fails to understand how it could work. 35-25 % chlorine atom equals to 4 - 6 electrons and based on the uncertainty of the AgCl usage, it might be wishful thinking to believe partial chloride anion in the cavity.

Response 3: In principle, the criticism of the reviewer is very much valid regarding the solubility of AgCl in DMSO. However, we can easily overcome the solubility problem when the host possesses high affinity for binding of chloride ion. There are experimental evidences in literature to support the present finding. The work of Clever et al and our own work published some time ago is a huge support.

The following paragraph regarding AgCl as source of Cl⁻ has been included in the ‘Discussion’ section of the revised manuscript.

“Notably, AgCl could be also used as a source of Cl⁻ ion. In fact, Clever *et al* has employed sparingly soluble AgCl as a source of Cl⁻ that displaced bound BF₄⁻ ion from the cavity of certain coordination cages so that AgCl got consumed and more soluble AgBF₄ remained in

solution.⁴⁷ We have also demonstrated the use of AgCl where the Cl⁻ ion displaced bound NO₃⁻ ion from the cavity of some coordination cages whereupon the more soluble AgNO₃ remained in solution.⁴⁸”

Query 4: The checkcif report are missing from the submission and there are some issues in the cif files, which should be at least commented or revised using the IUCR Output Validation Response Form, see <http://checkcif.iucr.org>

2c: Alert level B

PLAT084_ALERT_3_B High wR2 Value (i.e. > 0.25) 0.41 Report

PLAT342_ALERT_3_B Low Bond Precision on C-C Bonds 0.03 Ang.

3a: Alert level A

PLAT029_ALERT_3_A _diffn_measured_fraction_theta_full value Low . 0.893 Why?

4a: Alert level A

PLAT213_ALERT_2_A Atom C88 has ADP max/min Ratio 5.1 prolat

PLAT910_ALERT_3_A Missing # of FCF Reflection(s) Below Theta(Min). 53 Note

4a': Alert level B

PLAT342_ALERT_3_B Low Bond Precision on C-C Bonds 0.02539 Ang.

5c: Alert level A

THETM01_ALERT_3_A The value of sine(theta_max)/wavelength is less than 0.550

Calculated sin(theta_max)/wavelength = 0.5056

PLAT910_ALERT_3_A Missing # of FCF Reflection(s) Below Theta(Min). 70 Note

Alert level B

PLAT220_ALERT_2_B Non-Solvent Resd 1 C Ueq(max)/Ueq(min) Range 6.1 Ratio

PLAT241_ALERT_2_B High 'MainMol' Ueq as Compared to Neighbors of C6 Check

PLAT342_ALERT_3_B Low Bond Precision on C-C Bonds 0.02025 Ang.

6c: Alert level B

PLAT241_ALERT_2_B High 'MainMol' Ueq as Compared to Neighbors of C40 Check

PLAT910_ALERT_3_B Missing # of FCF Reflection(s) Below Theta(Min). 48 Note

Based on the above thgis referee can't support the publication of this work in Nature Communications. After careful revisions a proper journal could be Chemical Science (or Inorganic Chemistry).

Response 4: Checkcif reports for all the cifs have been uploaded along with the revised manuscript and supplementary information. Responses have been provided for the A and B level alerts.

REVIEWERS' COMMENTS:

Reviewer #3 (Remarks to the Author):

The author have improved the ms substantially and answered most of my concerns adquately. The ms can be published in its revised form.